# PAX2 and CAKUT Phenotypes: Report on Two New Variants and a Review of Mutations from the Leiden Open Variation Database

**DOI:** 10.3390/ijms24044165

**Published:** 2023-02-19

**Authors:** Susanna Negrisolo, Elisa Benetti

**Affiliations:** 1Laboratory of Immunopathology and Molecular Biology of the Kidney, Department of Women’s and Children’s Health, University of Padova, 35127 Padua, Italy; 2Pediatric Research Institute “IRP Città della Speranza”, 35127 Padua, Italy; 3Pediatric Nephrology, Department of Women’s and Children’s Health, Padua University Hospital, 35128 Padua, Italy

**Keywords:** *PAX2* gene, CAKUT, renal hypodysplasia, RHD, LOVD

## Abstract

*PAX2* is a transcription factor expressed during embryogenesis in the eye, ear, CNS, and genitourinary tract, and is one of the major regulators of kidney development. Mutations in this gene are associated with papillorenal syndrome (PAPRS), a genetic condition characterized by optic nerve dysplasia and renal hypo/dysplasia. In the last 28 years, many cohort studies and case reports highlighted *PAX2’s* involvement in a large spectrum of kidney malformations and diseases, with or without eye abnormalities, defining the phenotypes associated with *PAX2* variants as “*PAX2*-related disorders”. Here, we reported two new sequence variations and reviewed *PAX2* mutations annotated on the Leiden Open Variation Database 3.0. DNA was extracted from the peripheral blood of 53 pediatric patients with congenital abnormalities of the kidney and urinary tract (CAKUT). *PAX2* gene-coding exonic and flanking intronic regions were sequenced with Sanger technology. Two unrelated patients and two twins carrying one known and two unknown *PAX2* variations were observed. The frequency of *PAX2*-related disorders in this cohort was 5.8%, considering all CAKUT phenotypes (16.7% in the PAPRS phenotype and 2.5% in non-syndromic CAKUT). Although *PAX2* mutations have a higher frequency in patients with PAPRS or non-syndromic renal hypoplasia, from the review of variants reported to date in LOVD3, *PAX2*-related disorders are detected in pediatric patients with other CAKUT phenotypes. In our study, only one patient had a CAKUT without an ocular phenotype, but his twin had both renal and ocular involvement, confirming the extreme inter- and intrafamilial phenotypic variability.

## 1. Introduction

The *PAX2* gene (OMIM #167409) encodes for the paired-box transcription factor 2, a *PAX* tissue-specific gene family member, that is known to be involved in the development of the eye, ear, CNS, kidney, and urinary tract.

In human renal development, *PAX2* plays a key role at four weeks of gestation of metanephros patterning, with a strong expression in the Wolffian and Mullerian ducts, in the ureteral bud branching, and in the induction of the nephrogenic mesenchyme [1,2]. It is also expressed in the mesonephros, which is functional for a short time in the fetus. In developing metanephros, the expression of the *PAX2* and *PAX8* genes attenuate in the S-shaped stage before the nephrons are fully formed; in juvenile and adult kidneys, the expression of *PAX2* and *PAX8* are much lower than in the fetal period [3,4]. The critical role of the *PAX2* gene in tissues and organ development has been further supported by an in vivo analysis performed in a Pax2 knock-out mouse model. These studies have shown that homozygous deletions of Pax2 lead to early post-natal death with the absence of kidneys, ureters, and eyes [5].

In human nephrogenesis, *PAX2* expression is necessary for double doses (both alleles) for the correct development. Heterozygous mutations in this gene are associated with a wide renal disease pattern. *PAX2* gene mutations are characterized by inter- and intrafamilial phenotypic variability.

Different reports underlined an association between *PAX2* mutation and renal syndrome. For instance, it has been shown that *PAX2* haploinsufficiency is commonly found in patients with papillorenal syndrome (PAPRS, OMIM #120330). This is a genetic disorder characterized by optic nerve dysplasia, coloboma, and renal hypodysplasia, or oligomeganephronia, clinical features that dramatically enhance the risk of renal failure. PAPRS is often characterized by vesicoureteral reflux (25% of cases), hearing loss, malformations of the central nervous system, and joint and skin anomalies. Furthermore, other studies have shown that *PAX2* heterozygous variants are present in cases of isolated renal hypoplasia and other congenital anomalies of the kidney and urinary tract, including renal and ureteral anomalies with or without ocular abnormalities (CAKUT, OMIM #120330), and adult-onset focal segmental glomerulosclerosis (FSGS7, OMIM #616002) [6,7,8].

*PAX2* belongs to the *PAX* gene family, homeotic genes named after a highly conserved motif in the nucleotide sequence, called the paired box, which encodes an amino-terminal protein domain of about 128 amino acids (Prd) originally described in Drosophila. This domain has been observed in proteins that regulate the transcription of promoters for RNA polymerase II and are involved in embryogenesis. It can bind DNA and its correct functioning requires an octapeptide, a homeodomain, or a transactivation domain formed by a serine-rich sequence and threonine in the C terminal segment of the protein. These protein sequences can also interact with DNA targets [3]. *PAX* genes belong to the helix-turn-helix transcription factor superfamily and can function as both activator and repressor factors of transcription. They have been identified in a wide variety of species, including *C. elegans*, *Drosophila melanogaster*, *Danio rerio*, *Gallus gallus*, *Mus musculus*, and many mammalian species. In mice, there are nine members of the Pax family and there is high homology with the human *PAX* genes. Pax proteins and their alternatively spliced isoforms use different subdomains to bind DNA and thus modulate recognition specificity for different promoter sequences.

The *PAX* gene family is divided into four subgroups based on structural similarities, dependent on sequence homologies, and based on the presence of octapeptide, partial, or complete homeodomain domains:Subgroup I: includes PAX1 and PAX9, which contain the paired box and octapeptide domains.Subgroup II: includes PAX2, PAX5, and PAX8, which contain the paired-box, octapeptide, and partial homeodomain domains.Subgroup III: includes PAX3 and PAX7, which contain the paired-box, octapeptide, and total homeodomain domains.Subgroup IV: includes PAX4 and PAX6, which contain the paired-box and total homeodomain domains.

*PAX* genes are also defined as proto-oncogenes as their over-expression has been noted in various tumors. 

The human *PAX2* gene is located on human chromosome 10q24-25 and consists of 12 exons. Three exons (6, 10, and an alternative acceptor splice site located within exon 12) are alternatively spliced in humans, resulting in five different isoforms of transcripts, the function of which is not yet fully understood [3].

The most conserved exons (exons 2–4) of the PAX2 transcription factor encode the paired box domain, the N-terminal protein portion present in the II group of the *PAX* gene family. Other highly conserved exons are 5, 7, 8, and 9. They code for specific PAX2 domains. Exon 5 encodes for the octapeptide motif and exon 7 for the homeodomain. Exons 8 and 9 encode for the transactivation domain, a serine-threonine-rich sequence located in the C terminal segment of the protein. These domains are all important to make the protein fully functional (Figure 1). Although these domains are implicated in the regulatory activity of transcription factor, with DNA recognition and DNA-binding sequences, the paired box is the most involved and its two helix-turn-helix motifs are often mutational hotspots of the gene [9].

Referencing Salomon et al., 2001, we evaluated the presence of *PAX2* variants in patients with renal hypodysplasia, with or without eye abnormalities, enrolled starting in 2005 [10]. In 2007, we reported the first case of total deletion of *PAX2* in a patient with only renal phenotype [11]. In 2011, we carried out *PAX2* sequencing in a cohort of 20 pediatric patients who underwent renal transplantation and observed a frequency of mutations in isolated hypodysplasia of 10%. In that study, we also found two novel *PAX2* mutations in patients with no known ocular abnormalities and suggested *PAX2* mutation screening in all patients with CAKUT [6]. 

Here, we report the data obtained by a mutational screening of *PAX2* in pediatric CAKUT patients enrolled in our center.

## 2. Results

A total of 53 patients (35 males and 18 females, age 6–22 years) with syndromic (i.e., renal and urinary tract disease and ocular and/or ear features) and non-syndromic (i.e., renal and urinary tract phenotype only) CAKUT were recruited over three years. The observed phenotypes had mono- or bilateral renal hypo/dysplasia with or without other urinary tract abnormalities, bilateral or unilateral renal agenesis, oligomeganephronia, pyeloureteral duplicity, hydronephrosis, megaureter, and isolated VUR (Table 1). Twelve patients presented phenotypes that could be classified as PAPRS.

### Mutational Analysis in the Pediatric Cohort

Direct Sanger sequencing of *PAX2* coding regions highlighted three heterozygous variants in four patients: two unrelated patients and two twins (Table 2). All variations observed were localized in exon 2, one of the *PAX2* mutational hotspots: the known deletion c.69delC, a new missense mutation (c.212G>T), and a new indel (c.153_155delCTGinsTT). 

In this pediatric cohort, the frequency of the *PAX2* mutation was 5.8% (3/52) in all CAKUT phenotypes, whereas in non-syndromic CAKUT this was about 2.5%. Rather, the percentage of *PAX2* variations increased to 16.7% focusing on patients having PAPRS phenotype (RHD with ocular and/or ear phenotype). All these numbers were calculated considering that the c.153_155delCTGinsTT variation was found in two twins, thus counted only once (Table 3).

c.69delC is a *PAX2* mutation already reported by our group, resulting in a truncated protein of 29 amino acids [6]. This de novo mutation was identified in a girl with bilateral renal hypodysplasia, VUR, and hearing loss. The c.212G>T missense substitution was observed in a subject with a phenotype compatible with Renal-Coloboma syndrome. The patient had a chronic renal failure due to bilateral renal hypodysplasia, bilateral VUR, mild hypertrophy of the bladder neck, coloboma of the right eye, and a colobomatous dimple of the left eye. The de novo variant c.212G>T (p.Arg71Met) is located in a highly conserved base within the paired-box portion of all *PAX* genes (Figure 2). 

Polyphen2 predictive bioinformatics analysis defined this missense variant as “probably damaging”, and SIFT classified it as “deleterious”. The Esypred3D and SWISS-MODEL software obtained a three-dimensional model of the mutated PAX2 paired-box domain, exploiting the high sequence homology with the PAX5 paired-box domain, that had two deposited structures in the PDB protein database (PDB: 1K78, PDB: 1MDM). The ionic bonds existing for arginine 71 (Arg71) on the wild-type protein and those predicted by homology models for the mutated protein methionine 71 (Met71) were highlighted by the PyMol 3D visualization software. The conserved amino acid Arg71 is involved in DNA binding and plays an important structural role as it binds to the amino acids lysine 67 and the threonine 75 of the alpha helix (Figure 3). The Met71 variant is predicted to lose its DNA binding and acquire a new binding site in Glu74 in the alpha helix (Figure 3).

The indel variant c.153_155delCTGinsTT was observed in two monozygotic twins with renal cystic hypodysplasia, poor growth, and developmental delay (Figure 4). Only one of the twins also had colobomatous cysts and a congenital cataract in his right eye.

This sequence variant was not detected in the parents and it has never been reported in the mutational databases Human Gene Mutation Database (HGMD, http://www.hgmd.cf.ac.uk/ac, accessed on 1 March 2022), Clinvar database (https://www.ncbi.nlm.nih.gov/clinvar, accessed on 1 March 2022) and Leiden Open Variation Database (LOVD, https://databases.lovd.nl/shared/genes/PAX2, accessed on 1 March 2022). The prediction program Translate Tool (https://web.expasy.org/translate/, accessed on 1 March 2022) confirmed that the indel modifies the reading frame, introducing a stop codon after 31 amino acids from the variation p.Cys52Leufs*31 and generating a truncated protein of 83 amino acids (instead of 417), which is likely to be causative of the patient’s phenotype.

## 3. Discussion

Congenital anomalies of the kidney and urinary tract (CAKUT) are the leading cause of chronic renal failure in the pediatric population. These are malformations characterized by a wide phenotypic variability whose clinical relevance varies from less severe forms, with mild renal functional alterations, to severe forms, such as agenesis and renal hypodysplasia [12]. Studies in animal models suggest that these anomalies are due to a dysregulation of the complex nephrogenic program. Although causative mutations in humans have been identified in several genes involved in the development of the kidney and urinary tract, in most cases, etiopathogenesis remains to be clarified. The *PAX2* gene is involved in the development of the urinary tract from early metanephric induction to nephron patterning and differentiation. These phases are critical for the determination of most CAKUT phenotypes; therefore, *PAX2* is one of the most important candidate genes to be analyzed in a mutational screening.

One of the most completed collections of the human *PAX2* variant is the Leiden Open Variation Database (LOVD. https://databases.lovd.nl/shared/genes/PAX2, accessed on 31 December 2022), which was created in 2011. Before this, there was no actively maintained locus-specific database (LSDB) standardizing nomenclature and cataloging the extent of genetic variation in the *PAX2* gene and phenotypic variation in individuals. Since 2011, the mutations reported increased from 83 to 153 variations [13,14].

The database contains 349 entries comprising all public variants, identified by a nomenclature based on the NM_003990.3 transcript that encodes for the *e* isoform and NP_003981.2 PAX2 protein variant. The variants annotated are linked with the major genome browser (UCSC Genome Browser, Ensembl Genome Browser, and NCBI Sequence Viewer). Approximately 65% of variations were classified as pathogenic or likely pathogenic, and reported variants are localized in coding regions in 90% of cases and splice regions in the remaining 10%. Concerning the protein phenotypes, most pathogenic variations reported are missense (48% missense, 28% frameshift, 13% stop change, 6% inframe deletions or duplications, 1% no protein production, 1% silent, and 3% unknown variants). 

The *PAX2* mutational hotspots reported in the database are in exons 2, 3, and 4. Most of these mutations cause frameshifts, resulting in the formation of a truncated protein that is no longer able to bind DNA, such as c.76dup, the most frequently observed *PAX2* variant, reported in 59 records. There are a few variations affecting exons 5, 7, 8, and 9, which encode the octapeptide domain, the homeodomain, and the transactivation domains [13,15]. The associated variants phenotype is PAPRS in 52.8% of reports, 18% are CAKUT with or without ocular findings, 16% are RHD, 6.6% are FSGS, and 6.6% are ocular abnormalities ranging from myopia to nerve optic dysplasia without renal phenotype. The database also reports six chromosomic alterations: a balanced translocation (10:13) with a breakpoint on the *PAX2* locus in intron 3 or 4 (10q24.3q12.3), two large deletions encompassing 90 genes including *PAX2*, and other three deletions encompassing 2 to 44 genes including *PAX2* [11,16,17,18,19,20]. Although a complete deletion of the gene has been described in patients with renal phenotype only, *PAX2* mutations have never been reported in subjects with ocular phenotype only. 

In this study, the mutational screening of the *PAX2* gene in pediatric patients with the CAKUT phenotype revealed three gene sequence variations, two of which have never been reported before. The variations observed were coded and belong to missense and deletions protein phenotypes, similar to those in the LOVD3 PAX2 database. 

The deletion of c.69delC, the most frequently reported variation, observed in a patient with bilateral renal hypodysplasia, VUR, and hearing loss, caused the production of a truncated protein of 29 amino acids. The same mutation observed in a subject with the same malformation pattern was described for the first time in a previous study by our group [6]. Both patients did not have the ocular abnormalities typical of papillorenal Syndrome (PAPRS), thus confirming the association between mutations of the *PAX2* gene and non-syndromic pictures of hypodysplasia and VUR.

The c.212G>T missense substitution located in the exon 2 gene was observed in a patient with a PAPRS syndromic phenotype. The variant, not previously reported in the literature, resulted in the missense substitution p.Arg71Met in the paired-box domain of the protein and might link to the patient’s phenotype by bioinformatics analysis. In the literature, a missense mutation, with an amino acid change p.Arg71Thr, has been reported in the same codon in a pediatric patient with renal failure, nystagmus, coloboma, and polydactyly, and in an adult patient with malrotation, renal failure, and coloboma (both compatible with a syndromic pattern of PAPRS) [2,21]. The identification of this new mutation in our study confirms the high pathogenicity of the missense substitutions in this position of the paired box domain.

We identified a new indel variant, c.153_155delCTGinsTT, in monozygotic twins with renal cystic hypodysplasia, poor growth, and developmental delay. Only one of the twins presented congenital cataracts and coloboma of the right eye. The variant would create a frameshift leading to the replacement of the amino acid cysteine with the amino acid leucine in position 52 (p.Cys52Leufs*31). The final result was a 31 amino acid difference from the wild-type sequence before the premature termination of the protein. In the literature, there is only one other case of *PAX2* gene mutation (c.155G>A, p.Cys52Tyr) in monozygotic twins with discordant phenotypes [7]. One twin, who underwent kidney transplantation for chronic renal failure, presented unilateral dysplastic multicystic kidney and joint laxity. The other one had a normal renal function and no renal abnormalities but presented joint laxity and coloboma of the left eye. The authors concluded that the different clinical picture of monozygotic twins further confirms the extreme phenotypic variability of PAPRS syndrome. They hypothesized that environmental factors, modifying genes, and epigenetic factors may contribute to this variability. It is known that *PAX2* promotes histone methylation at kidney-specific loci during embryogenesis; therefore, this may impact developmental pathways. Furthermore, it is known that *PAX2*, as a transcription factor, may also act as a modifier in the expression of other genes in the renal developmental pathway [2]. For instance, a case reported by our group suggested that *PAX2* could act as a modifier gene in the Nail Patella phenotype. We reported a case of the co-presence of the *LMX1B* and *PAX2* variant in a girl with an extrarenal manifestation of Nail Patella Syndrome but also end-stage renal disease due to congenital bilateral renal hypodysplasia and vesicoureteral reflux. The *LMX1B* variant was *de novo*, whereas the *PAX2* variant was inherited from the mother, who had bilateral renal hypoplasia and mild chronic kidney disease. The molecular interaction between *LMX1B* and *PAX2* had already been reported in vitro and our findings suggested that the worst renal NPS phenotype of our patient could be due to the defective expression of the two genes during nephrogenesis [22].

## 4. Materials and Methods

Genetic analysis: All *PAX2* sequences were obtained by Sanger sequencing with PCR primers, as previously reported [6]. Patients’ and parents’ DNA was extracted according to the standard procedure from peripheral blood samples collected after written consent obtainment. The variants were named using the NM_003990.3 transcript reference sequence. For any identified variant, the pathogenicity prediction was established by in silico analysis by Polyphen2, SIFT, and Mutation Taster algorithms. The pathogenicity analysis of the variants was performed according to the ACMG standard and guidelines: all identified variants were classified as benign or likely benign, of uncertain significance, and pathogenic or likely pathogenic. Cases characterized by a correlation between variant pathogenicity and the patient’s phenotype were reported on the PAX2 LOVD3 database [13]. The alignment of the *PAX2* gene sequence in 17 species of eutherian mammals was performed using the open source ClustalW 2.1 multiple sequence alignment tool. The homology modeling prediction was performed by EasyPred3D and SWISS-MODEL bioinformatics tools. 3D structure visualization was obtained by PyMOL software.

## 5. Conclusions

The frequency of the *PAX2* mutation in this cohort was 5.8%, considering all CAKUT phenotypes reported. Only the renal hypo/dysplasia phenotypes, with ocular and/or ear phenotype (which can be included in PAPRS phenotype), was much higher (16.7%). The frequency of the *PAX2* mutation in non-syndromic CAKUT was about 2.5%. Our findings allow us to extend the spectrum of genotype–phenotype correlations in *PAX2*-related diseases, highlighting the relevance of screening the *PAX2* gene both in syndromic and isolated anomalies of kidney development. Our data also confirm that *PAX2* probably represents one of the important players in the determination of CAKUT in pediatric patients, suggesting that a targeted molecular diagnosis may lead to an earlier clinical molecular classification of the disease and a tailored work-up of the children with CAKUT.

## Figures and Tables

**Figure 1 ijms-24-04165-f001:**
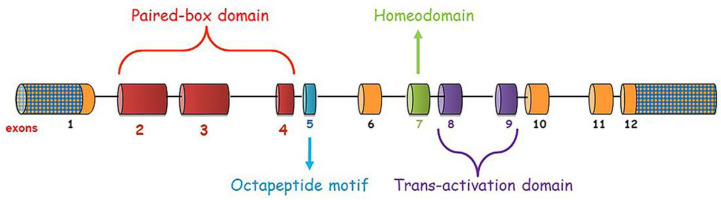
*PAX2* gene structure (exon–intron dimensions are not in scale measure). UTR regions are checkered.

**Figure 2 ijms-24-04165-f002:**
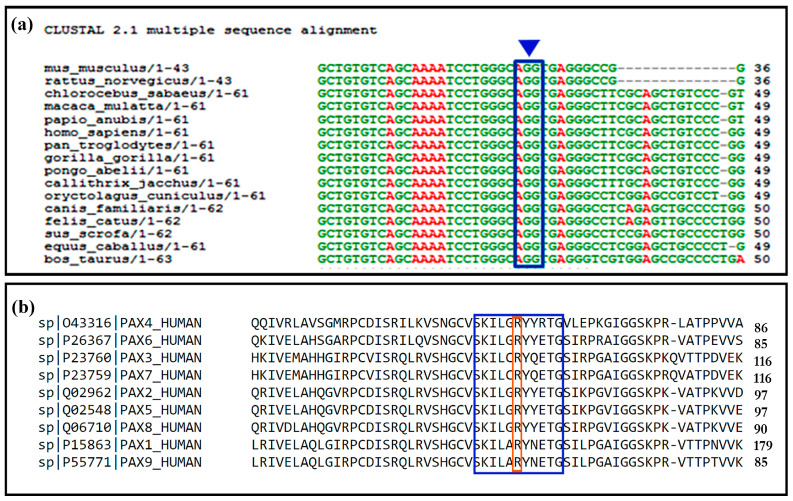
ClustalW output. (**a**) the alignment of the *PAX2* gene sequence in 17 species of eutherian mammals, in the blue box the codon wild type AGG, the arrow indicates the 212G nucleotide of c.212G>T variation. (**b**) the alignment of the PAX2 amino acid sequences relating to the paired domain of the nine proteins of the PAX family, in the blue box the alpha helics motif, in the red box the conserved R (Arg) of p.Arg72Met.

**Figure 3 ijms-24-04165-f003:**
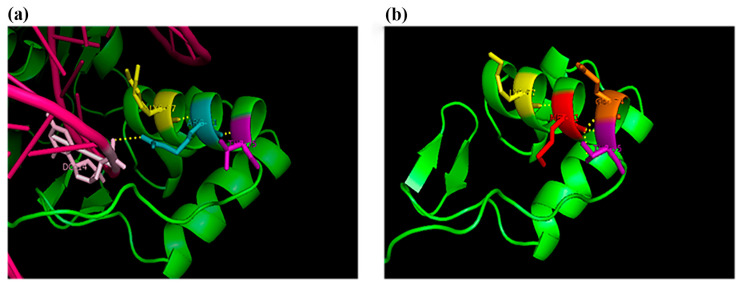
Detail of the 3D structures by PyMol visualization. (**a**) wild-type PAX5 paired-box domain with DNA interaction retrieved from PDB 1k78 structure: Arg71 (light blue) is the conserved amino acid; (**b**) 3D structure of PAX5 paired-box domain, elaborated by SWISS-MODEL prediction tool: the amino acid variant Met71 (magenta) seems to change the orientation of the alpha helix.

**Figure 4 ijms-24-04165-f004:**
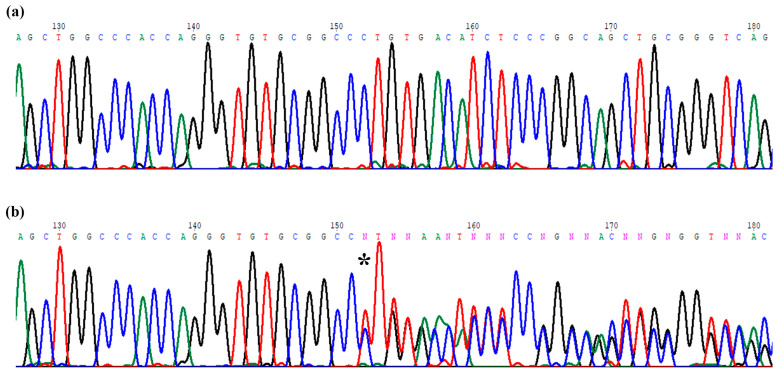
Chromatograms related to the sequence variation indel c.153_155delCTGinsTT. (**a**) the forward sequence of exon 2 wild type. (**b**) the forward sequence with the variations: the star (*) indicates the starting CTG nucleotides deletion and the TT insertion.

**Table 1 ijms-24-04165-t001:** Renal and ureteral phenotypes of the pediatric cohort.

CAKUT Phenotype	Total Cases	Bilateral	VUR Associated	With Other Urinary Abnormalities (PUV)	With Ocular and/or Ear Phenotype
RHD ^1^	34	26/34	17/34	4/34	12/34
Renal cystic dysplasia	2	-	-	-	1/2
Agenesis	9	-	2/9	-	-
Hydronephrosis	3	2/3	1/3	-	-
Megaureter	2	-	-	2/2	-
Isolated VUR	3	3/3	-	-	-

Acronymous legend: RHD: renal hypo/dysplasia; VUR: vesicoureteral reflux; PUV: posterior urethral valves. ^1^ RHD was the most represented phenotype in association with VUR and/or other urinary abnormalities (such as VUP and hydronephrosis).

**Table 2 ijms-24-04165-t002:** Variants highlighted in the four patients and the associated phenotypes.

	*PAX2* Variation	Exon	Predicted Protein Phenotype	Protein Type Variation	Renal Phenotype	Ocular and/or Ear Phenotype
P1	c.69delC	2	p.Val26Cysfs*2	frameshift	Renal hypodysplasia, VUR	yes
P2	c.212G>T	2	p.Arg71Met	missense	Chronic renal failure due to bilateral renal hypodysplasia, bilateral VUR, mild hypertrophy of the bladder neck	yes
P3	c.153_155delCTGinsTT	2	p.Cys52Leufs*31	frameshift	Renal cystic hypodysplasia, poor growth, and developmental delay	! no
P4	c.153_155delCTGinsTT	2	p.CysC52Leufs*31	frameshift	Renal cystic hypodysplasia, poor growth, and developmental delay	! yes

Acronymous legend: Patients: P1, P2, P3, P4. VUR: vesicoureteral reflux; ! these two patients were twins.

**Table 3 ijms-24-04165-t003:** *PAX2* variants distribution by renal phenotypes of the pediatric cohort.

Renal Phenotype	Cases Number	*PAX2* Variants in Unrelated Patients (Frequency)	Cases with Ocular and/or Ear Phenotype	*PAX2* Variants in Cases with Ocular and/or Ear Phenotype (Frequency)	*PAX2* Variants in Cases without Ocular and/or Ear Phenotype (Frequency)
RHD	34	2/34 (5.9%)	12	2/12 (16.7%)	0/22 (0%)
Other CAKUT	19 *	1/18 (5.6%)	1	1/1 (100%)	1/18 (5.6%)
Total CAKUT	53 *	3/52 (5.8%)	13	3/13 (23.1%)	1/40 (2.5%)

Legend: RHD: renal hypo/dysplasia; other CAKUT: comprised cystic dysplasia, agenesis, hydronephrosis, megaureter, isolates VUR. The * highlight the presence of two twins: the total number of unrelated patients tested was 52. Percentages are rounded to the first decimal place.

## Data Availability

The new variants reported in this study were submitted to LOVD3 PAX2 database: Variant #0000847240 (NC_000010.10:g.102509612_102509614delinsTT,*PAX2*(NM_003990.3):c.153_155delinsTT) https://databases.lovd.nl/shared/variants/0000847240#00025241, accessed on 31 December 2022. Variant #0000847239 (NC_000010.10:g.102509671G>T, *PAX2*(NM_003990.3):c.212G>T) https://databases.lovd.nl/shared/variants/0000847239#00025241, accessed on 31 December 2022.

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
