# Peer review of "PAX2 and CAKUT Phenotypes: Report on Two New Variants and a Review of Mutations from the Leiden Open Variation Database"

_ijms, 2023, doi:10.3390/ijms24044165_

Round 1
Reviewer 1 Report
In this article harmful mutations in PAX2 are described in the context of congenital abnormalities of kidney and urinary tract (CAKUT). These mutations were identified by sanger sequencing of DNA retrieved from 53 pediatric patients. Funnily, results to percentages of genotypes to phenotypes are written in the abstract only but not found in the result section. The manuscript needs reordering and language improvements. However, PAX2 variant detection and description seem to be valuable for the described disorders and hence aspects of this manuscript should be published.
Major:
- In the results section the text starts immediately with listing the 3 PAX2 mutations (ll134f). A guiding text of how these mutations are found is missing. Why are mutations selected on exon2 but not ex3 or ex4?
- provide table with all mutations found in PAX2, at least in supplement
- numbers of variants to phenotypes missing in the result section
- l199-l220: These paragraphs could be shifted to the result section. On the other hand: which conclusions are drawn from this? I don't see the value of this for your manuscript - what does the percentage of missense/frameshift/etc in variants mean? Maybe you could even consider to drop the figure 5 and associated text to this.
Minor:
- Figure 2: The red box hides the most important letter partially => probably R? Would be good to see c.212G>T and p.Arg72Met in the legend as well. An arrow to the mutation position 212G in the upper figure would also be helpful. Assigning the upper part as Figure 2a and the lower 2b also would benefit.
- PARPS or PAPRS? eg l14 and l52 => maybe better RCS?
- consistent writing would be helpful: Papillorenal syndrome or Papillorenal Syndrome should, eg l14 and l152
- References before or after the fullstop => be consistent, eg. l276 .[2]; l247 [6].
- writing: protein PAX2; gene PAX2 italicised (=> https://academic.oup.com/molehr/pages/Gene_And_Protein_Nomenclature?login=false)
- l73 C. elegance => C. elegans?
- l102: mutational hot spot => hot spots
- ll135f: Since these 2 variants are already listed in LOVD, please drop a note in the manuscript on this with corresponding IDs.
- l163: wild type PAX => PAX5
- l165: in deposited PDB 1k78 structure=> retrieved from PDB (ID: 1k78)
- l166: PAX => PAX5
- Figure 4: bad quality and in background unnecessary blurs seen; particularly position number hardly readable
- ll193f: Pax2 gene are involved => PAX2 is involved
- l206: referred to => based on?
- l207: e => an
- l211: o => or
- l213: As regards ?
- l224: hot spot => hot spots
- ll228f: such as those reported in some reports => leave out
- l229: In these reports PARPS/PAPRS
- l230: 6.6% => 6,6% (keep consistent either 6.6 or 6,6)
- l304: PAX2 gene => PAX2
- l287: .. => .
- l274: Actually => Currently?
- Last discussion paragraph needs rephrasing.
- l305: the most important => important
- l268: article title in italic => different to the other citations
Author Response
The manuscript has been sent to one of the English service providers suggested (MDPI’s Author Service) and edited by a native English writer.
We also reviewed our manuscript carefully and answered to the reviewers’ suggestions.
Issues to be addressed were the following:
Reviewer 1
Major review:
- In the results section the text starts immediately with listing the 3 PAX2 mutations (ll134f). A guiding text of how these mutations are found is missing. Why are mutations selected on exon2 but not ex3 or ex4?
We added a little guiding text of how the mutations are found.
The mutations were found by sequencing and their localization on exon 2 is not surprising, as is it a hotspot. We added the sentence “All variations observed were localized in exon 2, one of the PAX2 mutational hotspots (line138) in results.
- provide table with all mutations found in PAX2, at least in supplement
We provided a table of variants in the manuscript.
- numbers of variants to phenotypes missing in the result section
Number of variants has been added (l137).
- l199-l220: These paragraphs could be shifted to the result section. On the other hand: which conclusions are drawn from this? I don't see the value of this for your manuscript - what does the percentage of missense/frameshift/etc in variants mean? Maybe you could even consider to drop the figure 5 and associated text to this.
Figure 5 has been deleted as suggested. The LOVD3 PAX2 database overview section has not been moved to the results paragraph, as it enhances the discussion by providing insight into the knowledge of PAX2 variants. However, this section has been shortened and rearranged, in order to discuss our findings and to highlight the authors’ point of view.
Minor review:
- Figure 2: The red box hides the most important letter partially => probably R? Would be good to see c.212G>T and p.Arg72Met in the legend as well. An arrow to the mutation position 212G in the upper figure would also be helpful. Assigning the upper part as Figure 2a and the lower 2b also would benefit.
The red box has been shifted, so that the R amino acid can be read. c.212G>T and p.Arg72Met has been added to the legend, as well as “Figure 2a and 2b”. The name of the alignment tool “ClustalW” was added in the materials and methods section.
- PARPS or PAPRS? eg l14 and l52 => maybe better RCS?
We preferred PAPRS (l14 and l54).
- consistent writing would be helpful: Papillorenal syndrome or Papillorenal Syndrome should, eg l14 and l152
Corrected (l14 and l54)
- References before or after the fullstop => be consistent, eg. l276.[2]; l247 [6].
Corrected. We inserted references after fullstop.
- writing: protein PAX2; gene PAX2 italicised (=> https://academic.oup.com/molehr/pages/Gene_And_Protein_Nomenclature?login=false)
Corrected as suggested (PAX2).
- l73 C. elegance => C. elegans?
Corrected as suggested.
- l102: mutational hot spot => hot spots
Corrected as suggested.
- ll135f: Since these 2 variants are already listed in LOVD, please drop a note in the manuscript on this with corresponding IDs.
We added the LOVD ID assigned to the mutations in “Data Availability Statement”.
- l163: wild type PAX => PAX5
in the legends of figure 3 was added the number 5 to PAX paired box domain (l171)
- l165: in deposited PDB 1k78 structure=> retrieved from PDB (ID: 1k78)
Corrected as suggested (l173).
- Corrected as suggested (l171). l166: PAX => PAX5
- Figure 4: bad quality and in background unnecessary blurs seen; particularly position number hardly readable
Quality improved.
- ll193f: Pax2 gene are involved => PAX2 is involved
Corrected as suggested (l201-202)
- l206: referred to => based on?
Corrected as suggested (l214)
- l207: e => an
“e” => “e isoform” (l214)
- l211: o => or
Corrected as suggested (l217)
- l213: As regards ?
Corrected (l219)
- l224: hot spot => hotspots
Corrected as suggested (l224)
- ll228f: such as those reported in some reports => leave out
Corrected as suggested (l231)
- l229: In these reports PARPS/PAPRS
Corrected as suggested (l229)
- l230: 6.6% => 6,6% (keep consistent either 6.6 or 6,6)
Corrected as suggested (comma) l230
- l304: PAX2 gene => PAX2
Corrected as suggested
- l287: ..=> .
Corrected as suggested
- l274: Actually => Currently?
Edited by the English service.
- Last discussion paragraph needs rephrasing.
Revised.
- l305: the most important => important
Corrected as suggested (l314)
- l268: article title in italic => different to the other citations
Corrected as suggested.
Reviewer 2 Report
The authors raised a very interesting question of the PAX2 gene responsible for congenital abnormalities of kidney and urinary tract. This is very important for understanding development of kidney and urinary tract on molecular base together with possible genetic screening. The downside I see is that nowadays it would be more appropriate to use a larger genetic panel (exome sequencing using NGS) and add some functional experiments. Overall, the manuscript was clearly written and well-organized. However, there are several critical points to be noticed.
1. The language and writing need to be polished by a native speaker. Please check throughout the text, e. g.:
Page 1, line 19-21
PAX2 coding and flanking regions 19 sanger sequencing was performed in DNA from peripheral blood of 53 pediatric patients with 20 congenital abnormalities of kidney and urinary tract (CAKUT).
Page 2, line 52-55
First reports associated PAX2 haploinsufficiency with Papillorenal Syndrome 52 (PAPRS, OMIM #120330), a genetic disorder characterized by optic nerve dysplasia or 53 coloboma, and renal hypodysplasia or oligomeganephronia which predispose to renal 54 failure.
Please check throughout the text.
2. Figure 1 – Re-write in English “esoni” and add octapeptide motif rather than octapeptide. What was the reference for Figure 1?
3. Figure 4 – Use better quality of the chromatogram (resolution) and mark the position where the change occured.
4. What kind of mutations did you find (heterozygous vs. homozygous)? Please, specify.
5. Add “Author Contributions:” and fill “Data Availability Statement:” and “Acknowledgments:” or use “not applicable”.
6. The authors should unify the official gene name writing approved by the Human Genome Organization Gene Nomenclature Committee. Please check throughout the text.
7. Authors should be cautious with conclusions about altered protein expression. They have no concrete results for this.
8. It would be good to add a table with results of mutations found and phenotypes.
9. The section about database (LOVD) is too extensive. The reviewer strongly recommends the authors to reorganize this part to make their viewpoint more prominent.
Some of the minor points (Please check throughout the text):
Page 1 – In Affiliations, please, remove the word Affiliation
Page 1, line 21 – Capital letter in „two“.
Page 1, line 37 – four rather than 4 (number)
Please, write Latin words in italics (E. g. Page 2, line 73-74).
One sentence should not form a new paragraph.
Author Response
The manuscript has been sent to one of the English service providers suggested (MDPI’s Author Service) and edited by a native English writer.
We also reviewed our manuscript carefully and answered to the reviewers’ suggestions.
Issues to be addressed were the following:
Reviewer 2
- The language and writing need to be polished by a native speaker. Please check throughout the text, e. g.:
Page 1, line 19-21
PAX2 coding and flanking regions 19 sanger sequencing was performed in DNA from peripheral blood of 53 pediatric patients with 20 congenital abnormalities of kidney and urinary tract (CAKUT).
Page 2, line 52-55
First reports associated PAX2 haploinsufficiency with Papillorenal Syndrome 52 (PAPRS, OMIM #120330), a genetic disorder characterized by optic nerve dysplasia or 53 coloboma, and renal hypodysplasia or oligomeganephronia which predispose to renal 54 failure.
Please check throughout the text.
The manuscript has been sent to one of the English service providers suggested.
- Figure 1 – Re-write in English “esoni” and add octapeptide motif rather than octapeptide. What was the reference for Figure 1?
Corrected as suggested.
- Figure 4 – Use better quality of the chromatogram (resolution) and mark the position where the change occured.
Corrected as suggested: in detail we added the forward wt chromatogram and the mutation has been indicated with a star .
- What kind of mutations did you find (heterozygous vs. homozygous)? Please, specify.
Specified as requested (heterozygous mutations only) (line 137)
- Add “Author Contributions:” and fill “Data Availability Statement:” and “Acknowledgments:” or use “not applicable”.
We added Author Contributions, Data Availability and Acknowledgment.
- The authors should unify the official gene name writing approved by the Human Genome Organization Gene Nomenclature Committee. Please check throughout the text.
Checked.
- Authors should be cautious with conclusions about altered protein expression. They have no concrete results for this.
We tried to be more cautious with conclusions.
- It would be good to add a table with results of mutations found and phenotypes.
We added a table.
- The section about database (LOVD) is too extensive. The reviewer strongly recommends the authors to reorganize this part to make their viewpoint more prominent.
We reorganized the section.
Some of the minor points (Please check throughout the text):
Page 1 – In Affiliations, please, remove the word Affiliation
Deleted.
Page 1, line 21 – Capital letter in „two“.
Replaced
Page 1, line 37 – four rather than 4 (number)
Replaced
Please, write Latin words in italics (E. g. Page 2, line 73-74).
One sentence should not form a new paragraph.
Corrected.
Round 2
Reviewer 1 Report
Please provide a revised manuscript, where corrected passages are set in red colour. This helps the reviewing process and is quite common. To read your manuscript in the first form had been time consuming also because of many small errors. It would only be fair and respectful, to facilitate the reading of your revised version for the reviewer.
Author Response
We are sorry, the copy sent was the manuscript without the evidence of correction suggested by the reviewers and the English reviewer.
The newly uploaded manuscript has the corrected sentences set in red as requested (included English review).
The manuscript has been sent to one of the English service providers suggested (MDPI’s Author Service) and edited by a native English writer.
We also reviewed our manuscript carefully and answered to the reviewers’ suggestions.
Issues to be addressed were the following:
Reviewer 1
Major review:
- In the results section the text starts immediately with listing the 3 PAX2 mutations (ll134f). A guiding text of how these mutations are found is missing. Why are mutations selected on exon2 but not ex3 or ex4?
We added a little guiding text of how the mutations are found.
The mutations were found by sequencing and their localization on exon 2 is not surprising, as is it a hotspot. We added the sentence “All variations observed were localized in exon 2, one of the PAX2 mutational hotspots (line138) in results.
- provide table with all mutations found in PAX2, at least in supplement
We provided a table of variants in the manuscript.
- numbers of variants to phenotypes missing in the result section
Number of variants has been added (l137).
- l199-l220: These paragraphs could be shifted to the result section. On the other hand: which conclusions are drawn from this? I don't see the value of this for your manuscript - what does the percentage of missense/frameshift/etc in variants mean? Maybe you could even consider to drop the figure 5 and associated text to this.
Figure 5 has been deleted as suggested. The LOVD3 PAX2 database overview section has not been moved to the results paragraph, as it enhances the discussion by providing insight into the knowledge of PAX2 variants. However, this section has been shortened and rearranged, in order to discuss our findings and to highlight the authors’ point of view.
Minor review:
- Figure 2: The red box hides the most important letter partially => probably R? Would be good to see c.212G>T and p.Arg72Met in the legend as well. An arrow to the mutation position 212G in the upper figure would also be helpful. Assigning the upper part as Figure 2a and the lower 2b also would benefit.
The red box has been shifted, so that the R amino acid can be read. c.212G>T and p.Arg72Met has been added to the legend, as well as “Figure 2a and 2b”. The name of the alignment tool “ClustalW” was added in the materials and methods section.
- PARPS or PAPRS? eg l14 and l52 => maybe better RCS?
We preferred PAPRS (l14 and l54).
- consistent writing would be helpful: Papillorenal syndrome or Papillorenal Syndrome should, eg l14 and l152
Corrected (l14 and l54)
- References before or after the fullstop => be consistent, eg. l276.[2]; l247 [6].
Corrected. We inserted references after fullstop.
- writing: protein PAX2; gene PAX2 italicised (=> https://academic.oup.com/molehr/pages/Gene_And_Protein_Nomenclature?login=false)
Corrected as suggested (PAX2).
- l73 C. elegance => C. elegans?
Corrected as suggested.
- l102: mutational hot spot => hotspots
Corrected as suggested.
- ll135f: Since these 2 variants are already listed in LOVD, please drop a note in the manuscript on this with corresponding IDs.
We added the LOVD ID assigned to the mutations in “Data Availability Statement”.
- l163: wild type PAX => PAX5
in the legends of figure 3 was added the number 5 to PAX paired box domain (l171)
- l165: in deposited PDB 1k78 structure=> retrieved from PDB (ID: 1k78)
Corrected as suggested (l173).
- Corrected as suggested (l171). l166: PAX => PAX5
- Figure 4: bad quality and in background unnecessary blurs seen; particularly position number hardly readable
Quality improved.
- ll193f: Pax2 gene are involved => PAX2 is involved
Corrected as suggested (l201-202)
- l206: referred to => based on?
Corrected as suggested (l214)
- l207: e => an
“e” => “e isoform” (l214)
- l211: o => or
Corrected as suggested (l217)
- l213: As regards ?
Corrected (l219)
- l224: hot spot => hotspots
Corrected as suggested (l224)
- ll228f: such as those reported in some reports => leave out
Corrected as suggested (l231)
- l229: In these reports PARPS/PAPRS
Corrected as suggested (l229)
- l230: 6.6% => 6,6% (keep consistent either 6.6 or 6,6)
Corrected as suggested (comma) l230
- l304: PAX2 gene => PAX2
Corrected as suggested
- l287: ..=> .
Corrected as suggested
- l274: Actually => Currently?
Edited by the English service.
- Last discussion paragraph needs rephrasing.
Revised.
- l305: the most important => important
Corrected as suggested (l314)
- l268: article title in italic => different to the other citations
Corrected as suggested.
____________________________________________________________________________________
Reviewer 2
- The language and writing need to be polished by a native speaker. Please check throughout the text, e. g.:
Page 1, line 19-21
PAX2 coding and flanking regions 19 sanger sequencing was performed in DNA from peripheral blood of 53 pediatric patients with 20 congenital abnormalities of kidney and urinary tract (CAKUT).
Page 2, line 52-55
First reports associated PAX2 haploinsufficiency with Papillorenal Syndrome 52 (PAPRS, OMIM #120330), a genetic disorder characterized by optic nerve dysplasia or 53 coloboma, and renal hypodysplasia or oligomeganephronia which predispose to renal 54 failure.
Please check throughout the text.
The manuscript has been sent to one of the English service providers suggested.
- Figure 1 – Re-write in English “esoni” and add octapeptide motif rather than octapeptide. What was the reference for Figure 1?
Corrected as suggested.
- Figure 4 – Use better quality of the chromatogram (resolution) and mark the position where the change occured.
Corrected as suggested: in detail we added the forward wt chromatogram and the mutation has been indicated with a star .
- What kind of mutations did you find (heterozygous vs. homozygous)? Please, specify.
Specified as requested (heterozygous mutations only) (line 137)
- Add “Author Contributions:” and fill “Data Availability Statement:” and “Acknowledgments:” or use “not applicable”.
We added Author Contributions, Data Availability and Acknowledgment.
- The authors should unify the official gene name writing approved by the Human Genome Organization Gene Nomenclature Committee. Please check throughout the text.
Checked.
- Authors should be cautious with conclusions about altered protein expression. They have no concrete results for this.
We tried to be more cautious with conclusions.
- It would be good to add a table with results of mutations found and phenotypes.
We added a table.
- The section about database (LOVD) is too extensive. The reviewer strongly recommends the authors to reorganize this part to make their viewpoint more prominent.
We reorganized the section.
Some of the minor points (Please check throughout the text):
Page 1 – In Affiliations, please, remove the word Affiliation
Deleted.
Page 1, line 21 – Capital letter in „two“.
Replaced
Page 1, line 37 – four rather than 4 (number)
Replaced
Please, write Latin words in italics (E. g. Page 2, line 73-74).
One sentence should not form a new paragraph.
Corrected.
Reviewer 2 Report
I can consent to publication.
Author Response
We are sorry, the copy sent was the manuscript without the evidence of correction suggested by the reviewers and the English reviewer.
For accuracy we also send the manuscript with all correct sentences in red as requested (English proofreading included).
Round 3
Reviewer 1 Report
Dear authors,
thank you for sending a version with at least most of the changes to the first manuscript in red. The additional table helps to instantly catch the three describe variants, which you focus on in your manuscript.
Still, the abstract statements doesn't match entirely with the result section. Please check this carefully.
Major:
Although PAX2-related disorder percentages to phenotypes are given in the abstract and meanwhile in the conclusion, none of these percentages appear in the result section not to mention numbers on which these are based on. Which PAX2 variant contributes to which disorder? Related to that:
- Table 2: How many patients are affected by the given PAX2 variants? An additional column would be helpful with these numbers. Maybe from this, PAX2-related disorder percentages can be derived?
Alternatively, you could provide a table with the base numbers for each PAX2-related disorder and corresponding mutation numbers.
The bioinformatic analysis of PAX5, homolog of PAX2, suggests that the described variants might lead to the patient's phenotype, but this is not mechanistically shown. Therefore, this should be expressed in a more cautious way:
- l191-2: which is causative of the patient's phenotype => which is likely to be causative for...
- l253: appeared to be causative of => more carefully: might link to
Minor:
- Please check the writing: protein PAX2; gene PAX2 italicised, eg.:
l72, 85-89, 160, 161, 173: PAX => non-italic
- l201: PAX2 gene is involved... => either: The PAX2 gene is involved... or: PAX2 is involved...
- l207: of the PAX2 human variant => of the human PAX2 variants
- l218ff: Approximately 65% of variations were classified as pathogenic or likely
pathogenic reported variants are localized in coding regions in 90% of cases and splice regions in the remaining 10%. => sentence seems weird, probably better: ...likely pathogenic, and reported variants...
- l226: the most observed PAX2 variant => the most frequently observed...
- l278: PAX2,..., may act as a modifier gene in the... => PAX2, ..., may act as a modifier in the...
- l282/284: italic: de novo, in vitro
- l290: Patients ' => Patients'
Author Response
The newly uploaded manuscript has the revised sentences written in red (included English review).
Dear reviewer thank for you revisions. We corrected the manuscript as you required.
Here below (IN RED) our answers:
Major:
Although PAX2-related disorder percentages to phenotypes are given in the abstract and meanwhile in the conclusion, none of these percentages appear in the result section not to mention numbers on which these are based on. Which PAX2 variant contributes to which disorder? Related to that:
- Table 2: How many patients are affected by the given PAX2 variants? An additional column would be helpful with these numbers. Maybe from this, PAX2-related disorder percentages can be derived?
Alternatively, you could provide a table with the base numbers for each PAX2-related disorder and corresponding mutation numbers.
WE CORRECTED THE TABLE 1, WE INSERTED AN ADDITTIONAL COLUMN IN TABLE 2 AND A NEW SENTENCE L139.
IN THE RESULT SECTION WE ADDED A PARAGRAPH AND A FREQUENCIES TABLE (Table 3) L145-L154
The bioinformatic analysis of PAX5, homolog of PAX2, suggests that the described variants might lead to the patient's phenotype, but this is not mechanistically shown. Therefore, this should be expressed in a more cautious way:
- l191-2: which is causative of the patient's phenotype => which is likely to be causative for... CORRECTED
- l253: appeared to be causative of => more carefully: might link to…. CORRECTED
Minor:
- Please check the writing: protein PAX2; gene PAX2 italicised, eg.:l72, 85-89, 160, 161, 173: PAX => non-italic CORRECTED
- l201: PAX2 gene is involved... => either: The PAX2 gene is involved... or: PAX2 is involved... DONE
- l207: of the PAX2 human variant => of the human PAX2 variants DONE
- l218ff: Approximately 65% of variations were classified as pathogenic or likely pathogenic reported variants are localized in coding regions in 90% of cases and splice regions in the remaining 10%. => sentence seems weird, probably better: ...likely pathogenic, and reported variants... DONE
- l226: the most observed PAX2 variant => the most frequently observed...DONE
- l278: PAX2,..., may act as a modifier gene in the... => PAX2, ..., may act as a modifier in the... DONE
- l282/284: italic: de novo, in vitro DONE
- l290: Patients ' => Patients' DONE

Round 4
Reviewer 1 Report
Dear authors,
Thank you for providing the missing results for supporting your abstract. Good, that you have corrected the numbers accordingly.
However, please check your numbers in Table 3. It seems that 1/19 and 1/41 should be 1/18 and 1/40, respectively - 1/40* would help for understanding. Alternatively, you could think about including the twins in all numbers to avoid troubles with the percentages. This would need text adjustments.
There is no need to write your answers in capitol letters. We should stay professional.
Minor:
- l146: ..., whereas in non-syndromic CAKUT was about 2,4%. => whereas in non-syndromic CAKUT this was about 2,4%.
Author Response
Answer to review:
Dear Reviewer,
We apologize, we suppose there was a misunderstanding: the use of red and capital letters was only to highlight our responses within your review and make easy to distinguish them.
Thanks again for your suggestions. Changes are marked in red in the manuscript.
As for the numbers in Table 3, we corrected the data (1/18 and 1/40) accordingly. We agree that it becomes more intuitive.
However, in our paper it is more appropriate considering the twins as a single genetic event. Indeed, if we count them as two distinct patients, the PAX2 mutation frequency would increase from 5.8% to 7.5% in this cohort.
The symbol *, which highlights the presence of two twins, will remain only in the column “Cases number”.
Finally, in “l146” we corrected as you suggested: "...while in non-syndromic CAKUT this was about 2.5%".
Best regards.
